# Fine-tuning the pore environment of ultramicroporous three-dimensional covalent organic frameworks for efficient one-step ethylene purification

Yang Xie[1,5], Wenjing Wang [2,5], Zeyue Zhang [3,5], Jian Li [3,4], Bo Gui [1], Junliang Sun [3] ✉, Daqiang Yuan [2] ✉ & Cheng Wang [1] ✉

The construction of functional three-dimensional covalent organic frameworks (3D COFs) for gas separation, specifically for the efficient removal of ethane ($C_2H_6$) from ethylene ($C_2H_4$), is significant but challenging due to their similar physicochemical properties. In this study, we demonstrate fine-tuning the pore environment of ultramicroporous 3D COFs to achieve efficient one-step $C_2H_4$ purification. By choosing our previously reported 3D-TPB-COF-H as a reference material, we rationally design and synthesize an isostructural 3D COF (3D-TPP-COF) containing pyridine units. Impressively, compared with 3D-TPB-COF-H, 3D-TPP-COF exhibits both high $C_2H_6$ adsorption capacity (110.4 $cm^3$ $g^{-1}$ at 293 K and 1 bar) and good $C_2H_6/C_2H_4$ selectivity (1.8), due to the formation of additional C-H···N interactions between pyridine groups and $C_2H_6$. To our knowledge, this performance surpasses all other reported COFs and is even comparable to some benchmark porous materials. In addition, dynamic breakthrough experiments reveal that 3D-TPP-COF can be used as a robust absorbent to produce high-purity $C_2H_4$ directly from a $C_2H_6/C_2H_4$ mixture. This study provides important guidance for the rational design of 3D COFs for efficient gas separation.

Covalent organic frameworks (COFs) represent a new class of crystalline porous materials with two-dimensional (2D) or three-dimensional (3D) structures formed by the condensation of molecular building blocks[1–4]. Due to their high porosity, low density and high stability, COFs have found considerable potential in molecular adsorption and separation[5–9], catalysis[10–14], energy storage[15–18], optoelectronics[19–22], and membrane applications[6,23–25]. So far, most research efforts have focused on 2D COFs, and their applications have been extensively explored[26–30]. In contrast, research on 3D COFs has been progressed slowly, although their hierarchical nanopores and abundant accessible sites could make them particularly suitable for applications in gas adsorption and separation[31] as well as catalysis[32]. Fundamentally, the underdevelopment of 3D COFs is mainly due to challenges in obtaining highly crystalline samples and determining their crystal structures[33]. Additionally, designing the pore environment of 3D COFs for the target application is difficult, as sometime the steric hindrance of functional groups can complicate the structure determination process. For instance, we recently reported that by varying

[1]College of Chemistry and Molecular Sciences, Wuhan University, Wuhan 430072, China. [2]State Key Laboratory of Structural Chemistry, Fujian Institute of Research on the Structure of Matter, Chinese Academy of Sciences, Fuzhou 350002, China. [3]College of Chemistry and Molecular Engineering, Beijing National Laboratory for Molecular Sciences, Peking University, 100871 Beijing, China. [4]Department of Materials and Environmental Chemistry, Stockholm University, 10691 Stockholm, Sweden. [5]These authors contributed equally: Yang Xie, Wenjing Wang, Zeyue Zhang. ✉e-mail: junliang.sun@pku.edu.cn; ydq@fjirsm.ac.cn; chengwang@whu.edu.cn

the substituents from methoxy to phenyl, the topology of the designed 3D COFs changed from **pts** to **ljh**, which is not available in the ToposPro database and cannot be modeled from commonly used structural simulations[34]. The above structure uncertain may result in setting up wrong structure-property-function relationship, which will hinder the development of 3D COFs for target applications. Therefore, as a promising porous material, it is significant but challenging to construct functional 3D COFs with suitable pore environments for specific applications.

Ethylene ($C_2H_4$) is the most important crucial feedstock in the petrochemical industry, with a global production of ~200 million tons per year[35]. However, in order to remove the major byproduct ethane ($C_2H_6$), cryogenic distillation techniques are widely used in the industrial purification process of $C_2H_4$, consuming 0.3% of global energy[36]. To address this issue, adsorptive gas separation technique using porous adsorbents is considered to be effective in reducing costs and improving energy efficiency[37–40]. In particular, adsorbents that can be utilized for one-step purification of $C_2H_4$ have attracted significant interests due to the simplified separation process and energy savings of ~40%[41]. In principle, 3D COFs are a promising class of candidates for one-step $C_2H_4$ purification. Firstly, the pure organic nature of 3D COFs can provide a nonpolar pore environment, which facilitates the adsorption of $C_2H_6$, making one-step purification of $C_2H_4$ possible[42,43]. Secondly, the pore environments of 3D COFs can be adjusted by rational design of molecular building blocks, thereby achieving efficient $C_2H_6/C_2H_4$ separation[44,45]. Finally, the high stability brought by covalent linkages enables 3D COFs to meet the requirements of industrial reusability[46]. However, due to the above-mentioned issues, establishing a clear structure-property-function relationship for the design of 3D COFs with suitable pore environments to achieve efficient one-step $C_2H_4$ purification remains a significant challenge.

Herein, we report the fine-tuning of the pore environment of ultramicroporous 3D COFs for efficient one-step $C_2H_4$ purification. By choosing our reported 3D-TPB-COF-H as a reference material[47], we rationally designed and synthesized a functional 3D COF with pyridine units (3D-TPP-COF, Fig. 1). The continuous rotation electron diffraction (cRED) data showed that 3D-TPP-COF adopts a five-fold interpenetrated **pts** topology, which is isostructural to 3D-TPB-COF-H. Interestingly, 3D-TPP-COF exhibited significantly enhanced $C_2H_6$ capacity as well as higher $C_2H_6/C_2H_4$ selectivity compared to 3D-TPB-COF-H. To the best of our knowledge, this performance is superior to other reported COFs, and even comparable to some benchmark porous materials. In addition, dynamic breakthrough experiment indicated that 3D-TPP-COF can efficiently purify $C_2H_4$ from an equivalent molar mixture of $C_2H_6/C_2H_4$

in one step. Theoretical calculations suggest that the pyridine groups of 3D-TPP-COF can form additional C-H···N interactions with $C_2H_6$, thus enhancing the $C_2H_6$-trapping ability.

## Results

In order to optimize the pore environment of 3D COFs for efficient gas separation, we first synthesized the reference 3D-TPB-COF-H (Fig. 1) from 1,2,4,5-tetrakis-(4-formylphenyl)benzene (TPB-H) and tetra(p-aminophenyl)methane (TAPM) via [4 + 4] imine condensation reactions according to the literature[47]. Then, we rationally designed a new building block, namely 2,3,5,6-tetrakis-(4-formylphenyl)pyridine (TPP), to replace TPB-H as the quadrilateral precursor for the construction of a 3D COF (3D-TPP-COF, Fig. 1) with pyridine units. After screening various synthetic conditions, 3D-TPP-COF was successfully obtained as pale yellow powder by placing TPP and TAPM in chloroform and 12 M acetic acid aqueous solution (10:1, v/v) at 120 °C for 7 days. The atomic-level formation of 3D-TPP-COF was characterized by Fourier transform infrared (FT-IR) and solid-state nuclear magnetic resonance (ssNMR) spectroscopies. From the FT-IR spectrum (Supplementary Fig. 1), 3D-TPP-COF shows an intense peak at 1627 cm$^{-1}$, which corresponds to the stretching vibration band of imine bonds. In the ssNMR spectrum (Supplementary Fig. 2), a characteristic signal of carbon atom in the imine bonds was detected at 159 ppm. In addition, scanning electron microscopy experiments revealed that 3D-TPP-COF has a uniform octahedral morphology (Supplementary Fig. 3). Moreover, 3D-TPP-COF is thermally stable up to 550 °C under nitrogen atmosphere (Supplementary Fig. 5) and chemically stable in common solvents as well as under alkaline and weakly acidic conditions (Supplementary Fig. 6).

The crystallinity of 3D-TPP-COF was investigated by powder X-ray diffraction (PXRD) experiments. As shown in Fig. 2A, 3D-TPP-COF exhibited a series of intense diffraction peaks, indicating its long-range ordering. Subsequently, the cRED technique[48–50] was used to resolve the crystal structure of 3D-TPP-COF (Fig. 2B). The microcrystalline sample was first cooled down to 99 K and then several individual data sets were collected and processed, resulting unit cell parameters of $a = 28.42$ Å, $b = 7.17$ Å, $c = 28.12$ Å, $\alpha = \gamma = 90°$, $\beta = 97.21°$ with monoclinic symmetry. The reflection condition can be determined as $hkl$: $h + k = 2n$; $h0l$: $h = 2n$, $l = 2n$ from the 2D reciprocal slices, suggesting that the possible space group is $C2/c$ (No.15) or $Cc$ (No. 9). Based on the $hkl$ list from the cRED data, the initial structure model of 3D-TPP-COF including several structure fragments was solved with ShelxT in the space group of $C2/c$, and the integrated structure model containing the position of all non-hydrogen atoms was further completed using

**Fig. 1 | Chemical structure.** Schematic representation of the synthesis of 3D-TPP-COF and 3D-TPB-COF-H.

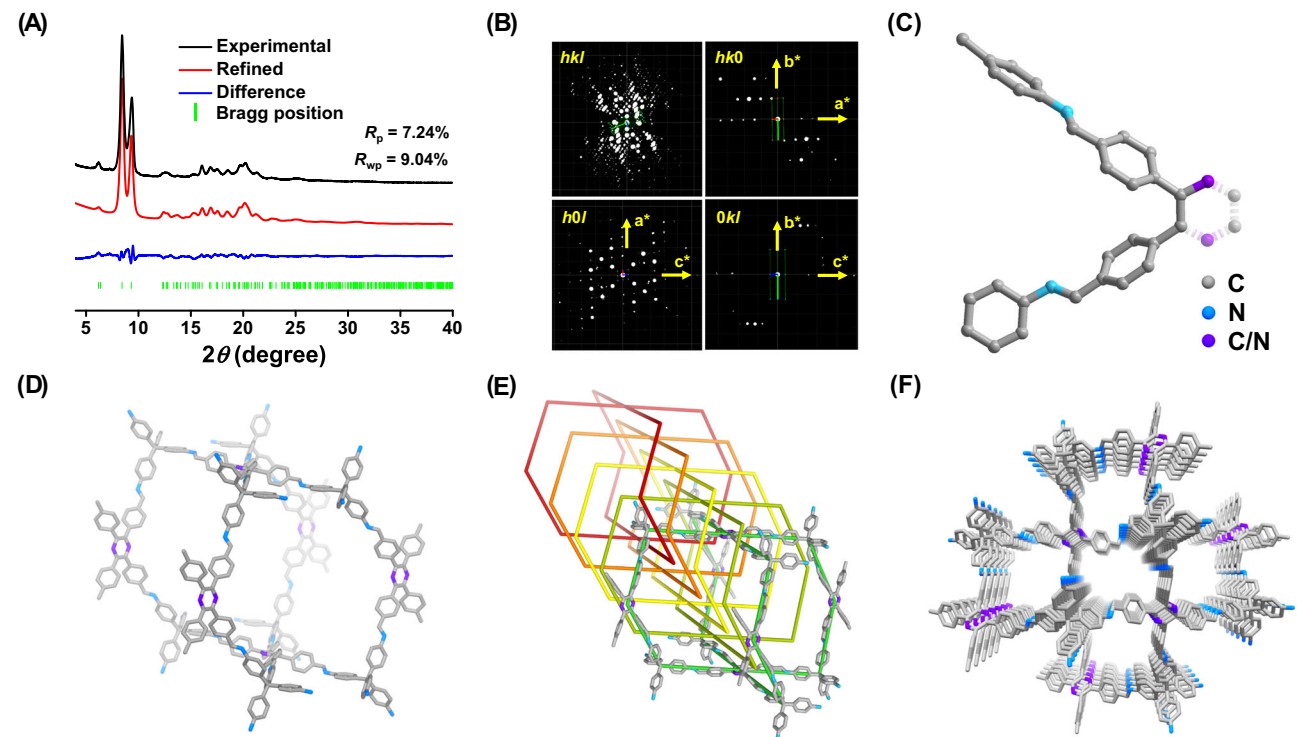

**Fig. 2 | Crystal structure of 3D-TPP-COF. A** PXRD patterns, **B** 3D reciprocal lattice and **C** asymmetric unit (containing whole aromatic rings) of 3D-TPP-COF. Structural representations of 3D-TPP-COF: **D** Single **pts** network, **E** the five-fold interpenetrated **pts** topology and **F** one-dimensional channels with exposed pyridine units.

the Material Studio software package. Since it is impossible to distinguish the C and N atoms at 1- and 4-positions of pyridine ring using cRED data, we finally adopted a structure model with identical probability of two pyridine orientations, where the C and N atoms were considered as a mixture and their occupancy rates were both 0.5 (Fig. 2C). The determined structure model was further refined by using Rietveld refinement against the experimental PXRD data (Supplementary Table 1), yielding a unit cell of $a = 28.691(9)$ Å, $b = 7.415(6)$ Å, $c = 27.857(5)$ Å, $\alpha = \gamma = 90°$, and $\beta = 95.651(4)°$ with $R_p$ of 7.24% and $R_{wp}$ of 9.04%. Finally, 3D-TPP-COF was determined to be a 5-fold interpenetrated pts structure, which is isostructural with 3D-TPB-COF-H but has exposed pyridine units in the pores (Fig. 2D–F).

The permanent porosity of two isostructural 3D COFs was determined by $N_2$ sorption measurement at 77 K. As shown in Supplementary Fig. 7, both 3D-TPP-COF and 3D-TPB-COF-H exhibited type I isotherms with a sharp increase at low pressure ($P/P_0 < 0.05$), indicating their microporous nature. The Brunauer-Emmett-Teller (BET) surface areas were calculated to be 1160 m$^2$ g$^{-1}$ and 1040 m$^2$ g$^{-1}$, respectively. According to density functional theory (DFT) method, both COFs have narrow pore size distributions centered at 0.52 nm. Given their high surface areas and ultramicroporous structures, we investigated the adsorption of both COFs on $C_2H_6$ and $C_2H_4$ at different temperatures (Fig. 3A and Supplementary Figs. 8 and 9). For 3D-TPB-COF-H, it has similar adsorption capacities for $C_2H_6$ (72.8 cm$^3$ g$^{-1}$) and $C_2H_4$ (72.3 cm$^3$ g$^{-1}$) at 293 K and 1 bar. In contrast, 3D-TPP-COF exhibited enhanced adsorption capacities for $C_2H_6$ (110.4 cm$^3$ g$^{-1}$) and $C_2H_4$ (105.3 cm$^3$ g$^{-1}$), with a corresponding increase of 51% and 45%. To the best of our knowledge, 3D-TPP-COF has the highest $C_2H_6$ adsorption capacity among reported $C_2H_6$-selective COFs (Fig. 3C)[51–54], even comparable to some benchmark porous materials such as ZJU-HOF-1[42], PCN-250[55] and UiO-67-NH$_2$[56].

Then the isosteric heat of adsorption ($Q_{st}$) values for 3D-TPP-COF and 3D-TPB-COF-H were calculated using the virial equation, based on isotherms recorded at different temperatures (Supplementary Figs. 14 and 15). At zero coverage, the $Q_{st}$ values of 3D-TPP-COF for

$C_2H_6$ and $C_2H_4$ were found to be 29.7 and 28.1 kJ mol$^{-1}$, respectively, while 3D-TPB-COF-H exhibited lower $Q_{st}$ values for $C_2H_6$ (26.8 kJ mol$^{-1}$) and $C_2H_4$ (25.3 kJ mol$^{-1}$). These results indicate that both COFs interact more strongly with $C_2H_6$ than with $C_2H_4$, and the introduction of pyridine groups can effectively enhance the host-guest interaction between COFs and gas molecules. Furthermore, the adsorption capacity of $C_2H_6$ is higher than that of $C_2H_4$ at low-pressure regions, suggesting that both COFs can be used as $C_2H_6$-trapping adsorbent for one-step $C_2H_4$ purification. To evaluate the separation potential, the ideal adsorbed solution theory (IAST) was applied to assess the selectivity of 3D-TPP-COF and 3D-TPB-COF-H for a 50/50 (v/v) $C_2H_6$/$C_2H_4$ gas mixture at 293 K. As can be seen in Fig. 3B, 3D-TPP-COF exhibits a $C_2H_6$/$C_2H_4$ selectivity of 1.8 at 1 bar, which is higher than that of 3D-TPB-COF-H (1.4) under the same conditions. Moreover, the $C_2H_6$/$C_2H_4$ selectivity of 3D-TPP-COF is higher than some top-performing $C_2H_6$-selective porous organic materials, including NKCOF-21[54] and CPOC-301[43]. These results clearly confirm that 3D-TPP-COF is a promising porous organic material with a good balance between high $C_2H_6$ uptake and $C_2H_6$/$C_2H_4$ selectivity.

Dynamic transient breakthrough experiments were carried out to evaluate the practical separation capacity of 3D-TPP-COF and 3D-TPB-COF-H on equimolar $C_2H_6$/$C_2H_4$ mixtures under ambient conditions. As shown in Fig. 3D, E, $C_2H_4$ elutes from the column first, while $C_2H_6$ broke through the column after a period delay. However, it should be emphasized that 3D-TPP-COF exhibits a longer separation time and better separation efficiency than 3D-TPB-COF-H, confirming again that the introduction of pyridine group in 3D COFs can efficiently improve the separation performance. To address the demand of practical industrial applications, multiple mixed gas breakthrough experiments were performed to test the recyclability of 3D-TPP-COF. No significant changes were observed in the separation performance within four continuous cycles (Fig. 3F), and the crystallinity of 3D-TPP-COF was well maintained after the breakthrough experiments (Supplementary Fig. 17). Therefore, by fine-tuning the pore environment of ultramicroporous 3D COFs, the rationally designed 3D-TPP-COF can be

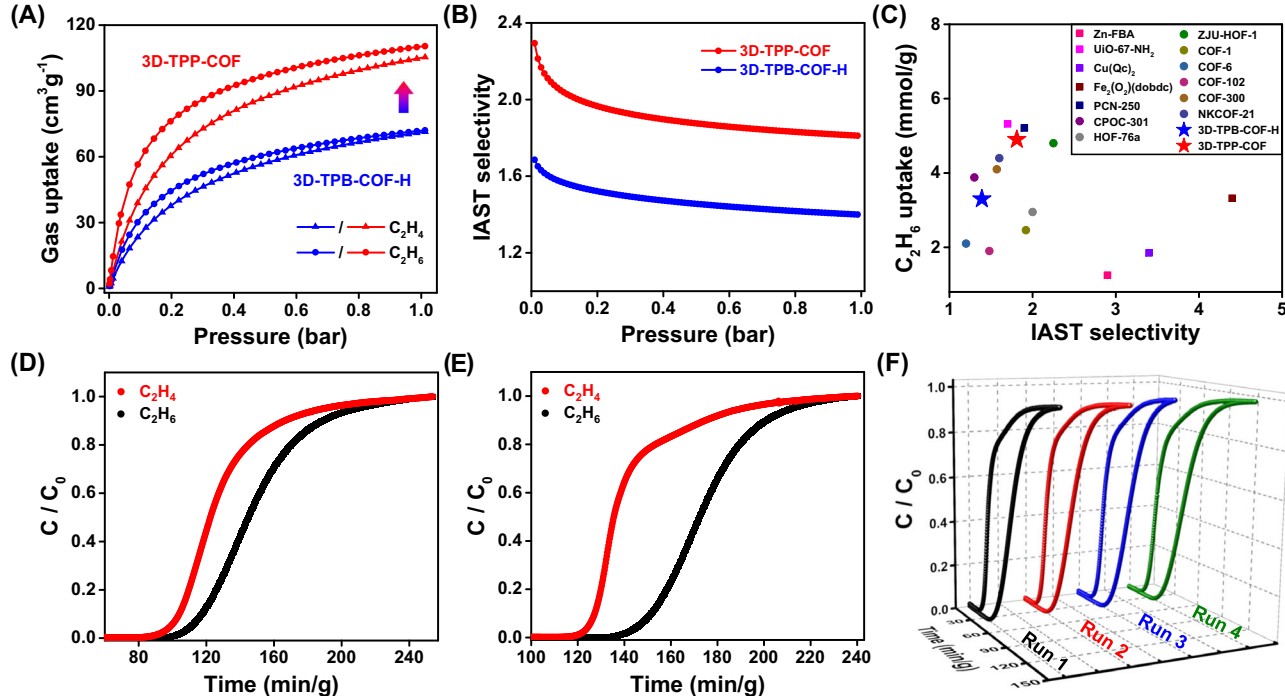

**Fig. 3 | Gas adsorption and separation study of 3D-TPP-COF and 3D-TPB-COF-H.** Gas adsorption and separation study of 3D-TPP-COF and 3D-TPB-COF-H. **A** Gas adsorption isotherms for 3D-TPP-COF and 3D-TPB-COF-H at 293 K. **B** The IAST selectivity of two 3D COFs for an equimolar $C_2H_6/C_2H_4$ mixture at 293 K. **C** Comparison of $C_2H_6$ uptake and IAST selectivity for 3D-TPP-COF with several top-performing $C_2H_6$-trapping porous materials at ambient condition. Experimental breakthrough curves for gas mixture of $C_2H_6/C_2H_4/He$ (10/10/80, v/v/v) over a packed column of 3D-TPB-COF-H (**D**) and 3D-TPP-COF (**E**) at 293 K and 1 bar. **F** The recyclability of 3D-TPP-COF under multiple mixed gas column breakthrough tests.

used for efficient one-step $C_2H_4$ purification, with high selectivity and robust stability.

To further elucidate the mechanism of efficient separation (Fig. 4), the interactions of 3D-TPP-COF and 3D-TPB-COF-H with gas molecules were calculated based on the simulated annealing algorithm and dispersion-corrected density functional theory (DFT-3D) using CP2K[42,57–59]. The random orientations of pyridine moieties in 3D-TPP-COF result in two configurations with different pore chemical environments, named as configuration-A and configuration-B (Supplementary Fig. 18). The primary binding sites for $C_2H_4$ and $C_2H_6$ molecules within the 3D COFs are summarized in Supplementary Fig. 19 and their corresponding binding energies were listed in Supplementary Table 3. As indicated in both binding sites and binding energies, 3D-TPB-COF-H exhibits less and weaker host-guest interactions with $C_2H_4$ and $C_2H_6$ molecules due to a less favorable pore environment, resulting in a weaker $C_2H_6$-trapping ability (Supplementary Table 4). In contrast, for 3D-TPP-COF, although the configuration-A similarly showed weak interactions with no obvious difference for $C_2H_6$ and $C_2H_4$ binding energies, the configuration-B shows more and stronger C−H···N interactions between the framework and $C_2H_6$ than those of $C_2H_4$ (−55.5 vs. −48.7 kJ mol⁻¹), as well as those of configuration-A. We thus speculated that such strong binding between the configuration-B of $C_2H_6$ is the cause for the selective adsorption of $C_2H_6$ over $C_2H_4$. Evidently, the modification of the pore environment with pyridine groups in 3D-TPP-COF enhances its one-step $C_2H_4$ purification performance significantly.

## Discussion
In summary, we have successfully designed and synthesized an ultra-microporous 3D COF with efficient one-step $C_2H_4$ purification capability by fine-tuning the pore environment. Compared with the reported 3D-TPB-COF-H, the rationally designed isostructural 3D-TPP-COF exhibits significantly enhanced $C_2H_6$ capacity and higher $C_2H_6$/

$C_2H_4$ selectivity, primarily due to the introduction of intrinsic pyridine moiety that forms additional interactions with the $C_2H_6$ guest. In addition, 3D-TPP-COF demonstrates robust stability and can efficiently capture $C_2H_6$ from $C_2H_6/C_2H_4$ mixture, leading to high-purity $C_2H_4$ production directly. This work provides a clear guidance for designing high-performance 3D COFs for $C_2H_4$ purification. Considering the promising applications of 3D COFs in gas separation, our work paves the way for developing other functionalized 3D COFs for efficient separation of industrially important gases. Further efforts are now underway in our laboratory.

## Methods
### Synthesis of 3D-TPP-COF
A Pyrex tube was charged with TAPM (22.8 mg, 0.06 mmol), TPP (29.7 mg, 0.06 mmol), chloroform (3.0 mL) and 12 M aqueous acetic acid (0.3 mL). After being degassed by freeze-pump-thaw technique for three times and then sealed under vacuum, the tube was placed in an oven at 120 °C for 7 days. The resulting precipitate was filtered, washed with DMF and dichloromethane for 2 days, dried at 120 °C under vacuum for 12 h. The activated 3D-TPP-COF was obtained as pale yellow powder (34.5 mg, 70% yield). Elemental analysis for the calculated: C, 84.85%; H, 4.67%; N, 8.53%. Found: C, 83.64%; H, 4.53%; N, 8.40%.

### Synthesis of 3D-TPB-COF-H
The 3D-TPB-COF-H was synthesized according to the literature[47]. A Pyrex tube was charged with TAPM (22.8 mg, 0.06 mmol), 1,2,4,5-Tetrakis-(4-formylphenyl) benzene (TPB-H) (29.7 mg, 0.06 mmol), chloroform (2.0 mL), n-butanol (0.1 mL) and 6 M aqueous acetic acid (0.2 mL). After being degassed by freeze-pump-thaw technique for three times and then sealed under vacuum, the tube was placed in an oven at 120 °C for 7 days. The resulting precipitate was filtered, washed with DMF and dichloromethane for 2 days, dried at 120 °C

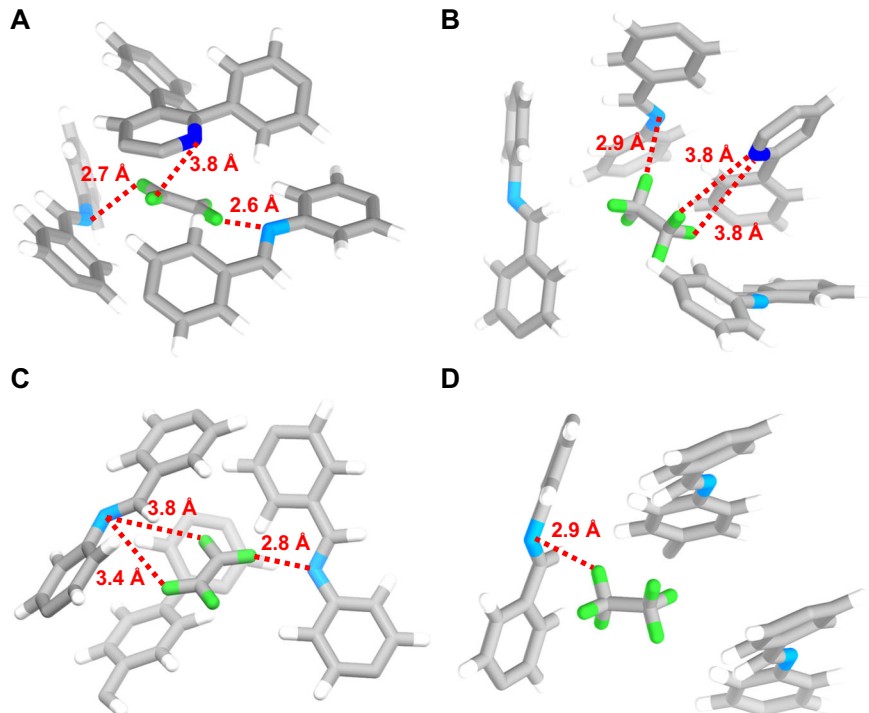

**Fig. 4 | Preferential gas adsorption sites in 3D-TPP-COF and 3D-TPB-COF-H.** Preferential $C_2H_4$ (**A**) and $C_2H_6$ (**B**) adsorption sites and close C-H⋯N interactions within 3D-TPP-COF. Preferential $C_2H_4$ (**C**) and $C_2H_6$ (**D**) adsorption sites and close C-H⋯N interactions within 3D-TPB-COF-H. For clarity, only one adsorbed gas molecule is shown in the ultramicropores. Nitrogen in pyridinyl is blue, nitrogen in imine is sky blue and hydrogen in C2 molecules is green.

under vacuum for 12 h. The activated 3D-TPB-COF-H was obtained as white powder (35.5 mg, 72% yield).

## Column breakthrough experiments

A home-built setup equipped with a mass spectrometer (Pfeiffer GSD320) was used to perform the mixed-gas breakthrough experiments. In a typical procedure, 0.31 g 3D-TPB-COF-H or 0.32 g 3D-TPP-COF powder was filled into a custom-made stainless-steel column (3.0 mm I.D. × 120 mm), which has the void space filled with silica wool, to perform breakthrough experiment with $C_2H_6/C_2H_4$/He (10:10:80, v/v/v) gas mixtures. The sample was activated by heating the packed column at 100 °C for 12 h under a constant He flow (10 mL min$^{-1}$ at 298 K and 1 bar). Then the He flow was turned off and the C2 hydrocarbon gas mixture was permitted to flow into the column (2 mL min$^{-1}$). The outlet effluent from the column was continuously monitored by the mass spectrometer. After breakthrough experiments, the column was heated at 100 °C for 12 h to regenerate the sample. The complete breakthrough of $C_2H_6$ was determined by using downstream gas composition reaching that of feed gas.

## Data availability

Crystallographic data for 3D-TPP-COF generated in this study have been deposited at the Cambridge Crystallographic Data Center, under deposition numbers CCDC 2300940. Copies of the data can be obtained free of charge via https://www.ccdc.cam.ac.uk/structures/. The data that support the findings within this study are available from the corresponding author upon request.

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

## Acknowledgements

C.W. gratefully acknowledges financial support from the National Natural Science Foundation of China (22225503 and U21A20285) and the Hubei Provincial Natural Science Foundation of China (2023AFA011). D.Y. acknowledges financial support from the National Natural Science Foundation of China (22275186). J.S. acknowledges financial support from the National Natural Science Foundation of China (22125102). We also thank the Core Facility of Wuhan University.

## Author contributions

Y.X. performed the synthesis and characterization of COFs, including NMR, PXRD, FT-IR and gas absorption. W.W. carried out $C_2H_4/C_2H_6$ adsorption and separation tests. D.Y. did the theoretical calculation of gas separation mechanism. Z.Z. and J.L. collected the electron diffraction data and solve the crystal structure of 3D-TPP-COF. B.G. helped to analyze the gas adsorption and separation results. C.W., D.Y. and J.S. designed and supervised the project. C.W. wrote the paper with the input of all authors.

## Competing interests

The authors declare no competing interests.
