## [Peer Review File · Nature Communications]

REVIEWER COMMENTS

Reviewer #1 (Remarks to the Author):

In this manuscript, Wang and co-authors report the synthesis of a new 3D-TPP-COF by modifying the pore environment with pyridine groups. The designed 3D-TPP-COF exhibited a significantly enhanced capacity for C₂H₆ and higher selectivity for C₂H₆/C₂H₄ compared to the 3D-TPB-COF-H. It is important that 3D-TPP-COF showed robust stability and can efficiently capture C₂H₆ from C₂H₆/C₂H₄ mixture. The results of theoretical calculations also suggest that the incorporation of pyridine groups into the pore environment of 3D-TPP-COF enhances its one-step C₂H₄ purification performance. In view of the insufficient and superficial structural characterizations of 3D COFs reported in some literatures, I appreciate the authors' efforts in analyzing the structure of 3D COFs and establishing a clear structure-property-function relationship to design 3D COFs with suitable pore environments for gas separation. In conclusion, this manuscript is important and well-organized and I recommend its acceptance after minor revisions. The following issues need to be addressed.

1. The unit of gas adsorption in the adsorption enthalpy curve should be consistent with that used in the corresponding adsorption isotherm.
2. The pore size obtained from DFT calculations should be compared with that derived from crystallographic data.
3. The authors solely focused on the adsorption of ethane and ethylene. What is the adsorption capacity of these 3D COFs for C₂H₂ gas? Is it feasible to achieve one-step purification of ethylene from a ternary mixture of C₂H₂/C₂H₄/C₂H₆?

Reviewer #2 (Remarks to the Author):

The authors reported the synthesis of a pyridine-containing 3D COF and its use for the separation of ethane/ethylene mixtures. Reasonable selectivity and capacity for the gas mixture were observed from this COF, which enables the demonstration of separation in breakthrough experiments. The authors compared the results to a 3D COF that was previously synthesized by them and concluded that the use of a pyridine moiety instead of a benzene moiety in one of the linkers has led to the improved separation efficiency, which is of apparent industrial relevance. The authors used cRED techniques to determine the 3D COF structure, and used modelling to reveal the COF-adsorbent interactions. Both technique sets have been previously employed and demonstrated in works from the authors, such as these reported in ref 48 and 44, respectively.

While the work reported here is of interest, it was felt as incremental compared to previous discoveries. Not surprisingly, the slight change of the atomic structure of one of the building blocks gave an

isostructural 3D COF as reported in ref 48. The gas separation effect is interesting though is on par with other reported organic porous materials and not the state-of-the-art overall. There also lacks some technical details (see below) and rigor that prevents the acceptance of this work in its current form.

Some detailed comments are listed below for the authors to consider:

The separation mechanism: The small pore size (0.5 nm) is close to the mean free path of the guest molecules (~0.42-0.44 nm). Under this condition, Knudsen diffusion becomes significant, the rate of which is inversely proportional to the square root of the molecular weight of the molecules. Since ethylene has a smaller kinetic diameter and lower molecular weight compared to ethane, it has a higher diffusion rate inside the pores compared to ethane. This aspect was not considered in this manuscript. Can the authors comment on this effect?

Capacity, kinetics, selectivity: the selectivity and capacity comparison chart isn't complete. The authors are suggested to refer to Figure 7 of the review article: *Ind. Eng. Chem. Res.* 2022, 61, 12269–12293. Placing the performance of the COF of interest in the overall grand scheme, the matrix is only mediocre. Also, kinetics is quite relevant for practical use, which was not commented in this work.

One-step separation: what is the purity of the separation product after the breakthrough? More details should be discussed regarding the breakthrough experiments, such as flow rate, use of a carrier gas, etc.

Molecular insight from binding energy calculations: the authors discussed two configurations A and B for the 3D COF. However, in reality, the pyridine N atom orientation is more likely being randomly distributed, as an average of configuration A and B, thus the modeling is not representative of the actual model. Given that A and B already showed different binding energetics, it raises the question whether the approach is appropriate. A related question is whether the modeling can sufficiently capture and describe the very weak van der Waals interactions (is the DFT by Materials Studio capable enough?). It becomes even more obscure when the authors ascribe the driving forces as C-H•••N and CH••• π interactions, since the C-H•••N distances of 3.8 and 4.1 Å, as shown in Figure 4, is way longer than any meaningful interaction distances.

Discussion on FT-IR: The authors stated that "From the FT-IR spectrum (Supplementary Figure 1), 3D-TPP-COF shows an intense peak at 1627 cm⁻¹, which corresponds to the stretching vibration band of imine bonds". The FTIR spectra of the starting material and the product are strikingly similar. I don't think the authors can draw conclusion about the imine formation based on the imine vibration given the spectrum overlap in this region. I have no doubt that the 3D imine COF was formed, however FT-IR probably isn't a good tool to differentiate that and the way it was written is misleading.

Reviewer #3 (Remarks to the Author):

This manuscript reported a study of a 3D COF and its potential separation for ethylene, and this beautiful work was done through efficient cooperation by three experts in the field of MOF/COF and crystal engineering. The crystalline structure for 3D TPP COF was clearly resolved by cRED technique, remarkably, 3D-TPP-COF exhibited high C₂H₄/C₂H₆ capacity and C₂H₆/C₂H₄ selectivity. I would support the publication of this manuscript in Nature Communications.

There are a few minor revisions and comments that I would encourage the authors to address before publication.

1: As far as I am aware of, most COFs show yellow or other obvious color, it is interesting that TPP COF is white in color, authors are suggested to discuss why it is white, and better provide the picture of TPP COF, and relevant spectral characterization like UV or PL can be also included in revision.

2: CIF files are suggested to be deposited into CCDC, which will be useful to the community, especially those who do not have the Materials Studio package.

3: It would be helpful to make comparisons of Q_{st} and adsorption selectivity for C₂H₆ and C₂H₄ toward other COFs/MOFs, a table of the comparison results can be listed in ESI to present an intuitive data visualization.

4: BET linear plot should be provided for TPP COF.

5: Typo and format. e.g. a needless letter d in the first page (behind the Email address) should be removed.

Point-By-Point Response to the Reviewers' Comments

Reviewer #1:

Comment 1: *In this manuscript, Wang and co-authors report the synthesis of a new 3D-TPP-COF by modifying the pore environment with pyridine groups. The designed 3D-TPP-COF exhibited a significantly enhanced capacity for C₂H₆ and higher selectivity for C₂H₆/C₂H₄ compared to the 3D-TPB-COF-H. It is important that 3D-TPP-COF showed robust stability and can efficiently capture C₂H₆ from C₂H₆/C₂H₄ mixture. The results of theoretical calculations also suggest that the incorporation of pyridine groups into the pore environment of 3D-TPP-COF enhances its one-step C₂H₄ purification performance. In view of the insufficient and superficial structural characterizations of 3D COFs reported in some literatures, I appreciate the authors' efforts in analyzing the structure of 3D COFs and establishing a clear structure-property-function relationship to design 3D COFs with suitable pore environments for gas separation. In conclusion, this manuscript is important and well-organized and I recommend its acceptance after minor revisions.*

Response: We thank the reviewer for these very positive comments and supporting on the publication.

Comment 2: *The following issues need to be addressed.*

1. *The unit of gas adsorption in the adsorption enthalpy curve should be consistent with that used in the corresponding adsorption isotherm.*

Response: We thank the reviewer for bringing this point to our attention. We have modified Supplementary Figures 14 and 15 to be consistent as suggested.

Supplementary Figure 14 The adsorption enthalpies (Q_{st}) of C₂H₆ (red) and C₂H₄ (blue) for 3D-TPP-COF.

Supplementary Figure 15 The adsorption enthalpies (Q_{st}) of C₂H₆ (red) and C₂H₄ (blue) for 3D-TPB-COF-H.

Comment 3: 2. The pore size obtained from DFT calculations should be compared with that derived from crystallographic data.

Response: We thank the reviewer for bringing this point to our attention. We have calculated the pore size distribution of 3D-TPP-COF by using Poreblazer (Molecular Simulation, 2011, 37, 1248) based on the crystal structure. As shown in Figure A, the simulated pore size distributions (centered at 0.46 nm) is quite close to the experimental data calculated (centered at 0.52 nm) from N₂ sorption isotherms.

Figure A Pore size distribution of 3D-TPP-COF by using Poreblazer based on the crystal structure.

Comment 4: 3. The authors solely focused on the adsorption of ethane and ethylene. What is the adsorption capacity of these 3D COFs for C₂H₂ gas? Is it feasible to achieve one-step purification of ethylene from a ternary mixture of C₂H₂/C₂H₄/C₂H₆?

Response: We thank the reviewer for bringing this point to our attention. We investigated the adsorption of acetylene by 3D-TPP-COF. As shown in Figure B, the acetylene adsorption capacity of 3D-TPP-COF is comparatively lower than ethane and

ethylene. Unfortunately, the breakthrough experiments showed that the one-step purification of ethylene from the mixture of $C_2H_2/C_2H_4/C_2H_6$ could not be achieved.

Figure B The gas adsorption of C₂ isomers by 3D-TPP-COF at 293 K (a) and experimental breakthrough curves for ternary mixture of $C_2H_2/C_2H_4/C_2H_6$ over a packed column of 3D-TPP-COF (b).

Reviewer #2:

Comment 1: The authors reported the synthesis of a pyridine-containing 3D COF and its use for the separation of ethane/ethylene mixtures. Reasonable selectivity and capacity for the gas mixture were observed from this COF, which enables the demonstration of separation in breakthrough experiments. The authors compared the results to a 3D COF that was previously synthesized by them and concluded that the use of a pyridine moiety instead of a benzene moiety in one of the linkers has led to the improved separation efficiency, which is of apparent industrial relevance. The authors used cRED techniques to determine the 3D COF structure, and used modelling to reveal the COF-adsorbent interactions.

Response: We thank the reviewer for these comments and their recognition of our work.

Comment 2: Both technique sets have been previously employed and demonstrated in works from the authors, such as these reported in ref 48 and 44, respectively. While the work reported here is of interest, it was felt as incremental compared to previous discoveries. Not surprisingly, the slight change of the atomic structure of one of the building blocks gave an isostructural 3D COF as reported in ref 48. The gas separation effect is interesting though is on par with other reported organic porous materials and not the state-of-the-art overall. There also lacks some technical details (see below) and rigor that prevents the acceptance of this work in its current form.

Response: We thank the reviewer for this comment, but we do not fully agree with the reviewer on this point. As a new kind of porous materials, the accurate structure determination of 3D COFs is very important to establish structure-property-function relationship that can guide the future design for targeted applications. However, it is very challenging to determine the structure of 3D COFs. In principle, the most direct

and powerful technique utilized is single-crystal X-ray diffraction (SCXRD). Unfortunately, the growth of large single crystals of 3D COFs suitable for SCXRD remains a very challenging task (e.g. *Science* **2018**, *361*, 48). In most cases, 3D COFs were obtained as polycrystalline powders, which is not suitable for SCXRD. A promising approach to resolve the crystal structure of polycrystalline powders is the use of recent newly developed continuous rotation electron diffraction (cRED) technique. In our previous report (Ref. 48), we designed and synthesized three isostructural polycrystalline 3D COFs and determined their structures using cRED technique. Impressively, the electron diffraction resolution of these three COFs reached up to 0.9–1.0 Å, which enables the direct localization of all non-hydrogen atoms from a polycrystalline COF sample for the first time. Therefore, the main point of this paper is to demonstrate **for the first time** the possibility to precisely characterize 3D COFs crystal structures at the atomic level using cRED technique, which can thus help in the future design of 3D COFs with tailored pore environments for targeted applications (e.g. gas adsorption and separation).

To be honest, it is very challenging to obtain 3D COFs with high crystallinity suitable for cRED technique. In most cases, people chose to simulate the crystal structure of 3D COFs based on PXRD patterns by following a topological approach. However, structural modelling of 3D COFs may become questionable due to the limited available information from PXRD data and the structural complexity. For instance, we recently reported that by varying the substituents from methoxy to phenyl, the topology of the designed 3D COFs changed from **pts** to an unprecedented **ljh**, which is not available in the ToposPro database and cannot be modeled from commonly used structural simulations (*J. Am. Chem. Soc.* **2021**, *143*, 7279). In another example, we reported a novel 3D COF adopts a unique **6×2-fold interpenetrated dia** topology (*J. Am. Chem. Soc.* **2023**, *145*, 11276), which is almost impossible to model by PXRD pattern. In fact, Prof. Omar Yaghi also suggested people use Retiveld refinement instead of the regular used Pawley refinement for the structure determination of 3D COFs, as the latter does not sufficiently support the correctness of the chosen unit cell and space group (*ACS Cent. Sci.* **2020**, *6*, 1255). Therefore, structural modeling of 3D COFs should be conducted with extreme care (*Acc. Chem. Res.* **2020**, *53*, 2225; *ACS Cent. Sci.* **2020**, *6*, 1255; *Trends Chem.* **2022**, *4*, 437).

Recently, the investigation of 3D COFs in gas separation has attracted attentions (e.g., *J. Am. Chem. Soc.* **2022**, *144*, 5643; *J. Am. Chem. Soc.* **2022**, *144*, 23081; *Angew. Chem. Int. Ed.* **2022**, *61* e202204899). However, most of these COFs structures was simulated based on PXRD patterns, not determined by suggested 3D ED or SCXRD technique. Considering the promising applications of 3D COFs in gas separation and the possibility that more people will may work on this field, there is an urgent need to establish a clear structure-property-function relationship for designing 3D COFs with suitable pore environments for gas separation. Therefore, we design this research and report the fine-tuning pore environment to construct 3D COFs for efficient one-step C₂H₄ purification. The structure of 3D COF was clearly characterized by cRED technique, and the separation performance is better than all other reported results in COFs and even comparable to some benchmark porous materials. According to the

accurate structure determination, the further study clearly revealed that the incorporation of pyridine unit effectively increases C₂H₆ adsorption capacity and C₂H₆/C₂H₄ selectivity, due to the formation of additional C-H···N interactions between pyridine groups and C₂H₆. We strongly believe our work provide a standard design for developing other functionalized 3D COFs for efficient separation of C₂H₆/C₂H₄, including other industrially important gases in future.

Regarding to the concerns from the reviewer about technical details and rigor, we have been clearly revised point by point (please see below) and we hope the revised manuscript can be accepted in the esteemed journal of *Nature Communications*.

Comment 3: *Some detailed comments are listed below for the authors to consider: The separation mechanism: The small pore size (0.5 nm) is close to the mean free path of the guest molecules (~0.42-0.44 nm). Under this condition, Knudsen diffusion becomes significant, the rate of which is inversely proportional to the square root of the molecular weight of the molecules. Since ethylene has a smaller kinetic diameter and lower molecular weight compared to ethane, it has a higher diffusion rate inside the pores compared to ethane. This aspect was not considered in this manuscript. Can the authors comment on this effect?*

Response: We thank the reviewer for bringing this point to our attention. We agree with the reviewer that Knudsen diffusion can indeed exert an influence on separation performance, particularly when the pore size is close to the mean free path of the guest molecules. Nevertheless, in our study, as the structural characteristics of the two 3D COF cavities are nearly identical, the Knudsen diffusion may have similar effect on gases in two 3D COFs and not be the primary factor contributing to the distinctions observed in the separation performance. Based on our studies and calculations, the contrasting separation performances primarily stem from variations in host-guest interactions. Therefore, this aspect was not considered in the manuscript.

Comment 4: *Capacity, kinetics, selectivity: the selectivity and capacity comparison chart isn't complete. The authors are suggested to refer to Figure 7 of the review article: Ind. Eng. Chem. Res. 2022, 61, 12269–12293. Placing the performance of the COF of interest in the overall grand scheme, the matrix is only mediocre. Also, kinetics is quite relevant for practical use, which was not commented in this work.*

Response: We thank the reviewer for bringing this point to our attention. We have revised the comparison chart (Supplementary Figure 16) with enriched data according to the suggestion. For kinetics, we agree with the reviewer that it is quite relevant for practical use. In the gas adsorption process of the examined system, equilibrium was expeditiously attained with no discernible kinetic effect. However, the emphasis in this work is to set up a clear guidance for designing high-performance 3D COFs for ethylene purification. The single-component gas adsorption and dynamic gas breakthrough experiments have proved the availability of performance enhancement through fine-tuning the pore environment of 3D COFs. Consequently, the kinetics effect in this work was not considered. The further research about the design and synthesis of 3D COFs for practical use is ongoing in our lab.

Supplementary Figure 16 Comparison of C₂H₆ capacity and C₂H₆/C₂H₄ selectivity for porous C₂H₆-selective adsorbents at ambient condition.¹⁰ Selectivity values refer to IAST selectivity for a 50/50 C₂H₆/C₂H₄ mixture.

Comment 5: *One-step separation: what is the purity of the separation product after the breakthrough? More details should be discussed regarding the breakthrough experiments, such as flow rate, use of a carrier gas, etc.*

Response: We thank the reviewer for bringing this point to our attention. The purity of produced ethylene reaches 99.8% in one-step separation process by using 3D-TPP-COF as adsorbent, which is higher than that of 98.8% by 3D-TPB-COF-H. The details about the breakthrough experiments can be found in Supplementary Section 4.

Comment 6: *Molecular insight from binding energy calculations: the authors discussed two configurations A and B for the 3D COF. However, in reality, the pyridine N atom orientation is more likely being randomly distributed, as an average of configuration A and B, thus the modeling is not representative of the actual model. Given that A and B already showed different binding energetics, it raises the question whether the approach is appropriate. A related question is whether the modeling can sufficiently capture and describe the very weak van der Waals interactions (is the DFT by Materials Studio capable enough?). It becomes even more obscure when the authors ascribe the driving forces as C-H...N and CH... π interactions, since the C-H...N distances of 3.8 and 4.1 Å, as shown in Figure 4, is way longer than any meaningful interaction distances.*

Response: We thank the reviewer for this insightful comment. Regarding the discussion of the configurations for 3D-TPP-COF, we acknowledge that the pyridine N atom orientation is likely to be randomly distributed in reality, and taking the average of configuration A and B would better represent the actual model. However, with current single-crystal diffraction techniques, even electron diffraction single-crystal diffraction, it is still hard to distinguish this statistical distribution of pyridine N atom in the 3D-TPP-COF. Considering that the N atom only locates on 1- or 4- position in

the pyridine ring, there could be three situations with 0, 1, or 2 kinds of pyridine N along the channel of 3D-TPP-COF (Figure C). Therefore, two typical configurations were constructed to simulate the actual structure and used to explain the subsequent gas adsorption properties. Although this approximates the structure, we believe this can also help us explain the gas adsorption properties of the actual structure to some extent. According to our calculation results, the binding energetic difference of two gases within configuration A is close to 0 and much smaller than that within configuration B, which means that configuration B may play a major role in the gas adsorption process of 3D-TPP-COF. Thus, although the actual model may statistically take the average of two configurations, configuration B is still the leading cause for the overall ethane-selective adsorption behavior of 3D-TPP-COF. As for the calculating method, to more accurately estimate the static binding energies, we carried out standard density functional theory (DFT) implemented dispersion corrections using DFT-D3 method as implemented in the software package CP2K-2023.2. The dispersion corrections using DFT-D3 method have been proven to be effective in explaining the separation mechanism by some influential literature (*J. Am. Chem. Soc.* **2019**, *141*, 5014; *Nat. Commun.* **2021**, *12*, 197), instead of previous DFT-D2 method to describe the weak interactions. Moreover, the newly calculated results show an increasing binding energetic difference of two gases within configuration B compared to the old one (6.8 vs 1.9 kJ/mol). We have revised the related figures, tables, and expressions in the manuscript and supplementary information based on the new calculation result and reviewer's suggestion.

Figure C Different types of pyridinyl nitrogen in channels of 3D-TPP-COF. For clarity, the pyridinyl nitrogen and imine nitrogen are shown in blue and sky blue, respectively.

Figure 4 | Preferred gas adsorption sites in 3D-TPP-COF and 3D-TPB-COF-H. Preferred C₂H₄ (A) and C₂H₆ (B) adsorption sites and close C-H···N interactions within 3D-TPP-COF. Preferred C₂H₄ (C) and C₂H₆ (D) adsorption sites and close C-H···N interactions within 3D-TPB-COF-H. For clarity, only one adsorbed gas molecule is shown in the ultramicropores. Nitrogen in pyridinyl is blue, nitrogen in imine is sky blue and hydrogen in C₂ molecules is green.

Supplementary Figure 19 Calculated first adsorption sites of C₂ gases in 3D COFs. C₂H₄ in 3D-TPB-COF-H (a) and 3D-TPP-COF [configuration-A (b) and configuration-B (c)]; C₂H₆ in 3D-TPB-COF-H (d) and 3D-TPP-COF [configuration-A (e) and configuration-B (f)].

Supplementary Table 3 The binding energies (ΔE) for C2@3D COFs calculated by the Dmol3 module.

	3D-TPB-COF-H	3D-TPP-COF	
		configuration-A	configuration-B
E_{COF} (ha)	-1591.9790	-1606.5995	-1606.5829
$E_{\text{C}_2\text{H}_4}$ (ha)	-13.7226		
$E_{\text{C}_2\text{H}_4@\text{COF}}$ (ha)	-1605.7177	-1620.3392	-1620.3240
$\Delta E_{\text{C}_2\text{H}_4@\text{COF}}$ (ha)	-0.0161	-0.0171	-0.0185
$\Delta E_{\text{C}_2\text{H}_4@\text{COF}}$ (kJ/mol)	-42.22	-44.79	-48.65
$E_{\text{C}_2\text{H}_6}$ (ha)	-14.9543		
$E_{\text{C}_2\text{H}_6@\text{COF}}$ (ha)	-1606.9503	-1621.5706	-1621.5582
$\Delta E_{\text{C}_2\text{H}_6@\text{COF}}$ (ha)	-0.0170	-0.0168	-0.0210
$\Delta E_{\text{C}_2\text{H}_6@\text{COF}}$ (kJ/mol)	-44.70	-44.19	-55.46

Supplementary Table 4 The interactions between the gas molecules and host framework in 3D-TPB-COF-H and 3D-TPP-COF-B.

		3D-TPB-COF-H		3D-TPP-COF	
		H \cdots A (Å)		H \cdots A (Å)	
C ₂ H ₄	C-H \cdots N	2.80	3.37	2.55	3.84
		3.76	3.98	2.68	4.00
				4.23	
	C-H \cdots π	3.64	3.71	3.22	3.71
		4.14	4.17	4.13	4.19
		4.34	4.47	4.33	
C ₂ H ₆	C-H \cdots N	2.88	3.74	2.90	3.81
		3.88	3.98	3.83	4.15
		4.09		4.18	4.45
	C-H \cdots π			4.46	
		3.34	3.65	3.69	3.97
		4.09	4.17	3.74	4.01
	4.23	4.45			

Page 10: To further elucidate the mechanism of efficient separation, the interactions of 3D-TPP-COF and 3D-TPB-COF-H with gas molecules were calculated based on the simulated annealing algorithm and dispersion-corrected density functional theory (DFT-3D) using CP2K.^{43, 58-60}

Page 10: In contrast, for 3D-TPP-COF, although the configuration-A similarly showed weak interactions with no obvious difference for C₂H₆ and C₂H₄ binding energies, the configuration-B shows more and stronger C–H···N interactions between the framework and C₂H₆ than those of C₂H₄ (-55.5 vs -48.7 kJ mol⁻¹), as well as those of configuration-A.

Supplementary Section 5: The free 3D-TPP-COF, C₂ hydrocarbon guests, and C2@3D-TPP-COF were further optimized by DFT-D3 calculations performed in CP2K¹³ with the PBE functional,¹⁴ DZVP-MOLOPT-SR-GTH basis sets,¹⁵ GTH-PBE pseudopotential,¹⁶ a plane wave grid cutoff of 400 Ry, and the Grimme-D3 dispersion correction.¹⁷ The optimizations converged when the energy difference was less than 1.0×10^{-6} a.u. The inner SCF convergence for each optimization step followed the same criteria along with CP2K defaults for root mean square and max displacement criteria. The binding energies (ΔE_{bind} in kJ mol⁻¹) were calculated as the differences in total energies E between fully optimized C2@3D-TPP-COF and the 3D-TPP-COF and C₂ hydrocarbon guests in terms of the following equation:

Comment 7: Discussion on FT-IR: The authors stated that “From the FT-IR spectrum (Supplementary Figure 1), 3D-TPP-COF shows an intense peak at 1627 cm⁻¹, which corresponds to the stretching vibration band of imine bonds”. The FTIR spectra of the starting material and the product are strikingly similar. I don't think the authors can draw conclusion about the imine formation based on the imine vibration given the spectrum overlap in this region. I have no doubt that the 3D imine COF was formed, however FT-IR probably isn't a good tool to differentiate that and the way it was written is misleading.

Response: We thank the reviewer for bringing this point to our attention. To verify the construction of imine-linked COF, it is important to characterize the formation of imine bonds. Due to the full range display in FT-IR spectra, there seems to be a spectrum overlap in imine vibration region. To help the reader clearly see the difference between the starting material and product, we added an enlarged FT-IR spectra in the revised supplementary information. As shown in Supplementary Figure 1, there is an absorption peak at 1627 cm⁻¹ (green dashed line) for both COFs (black and red curves), which clearly demonstrates the formation of imine bonds. Besides, the ¹³C solid-state NMR spectrum of 3D-TPP-COF (Supplementary Figure 2) clearly confirm the formation of imine bond.

Supplementary Figure 1 Full (a) and enlarged (b) FT-IR spectra of 3D-TPB-COF-H (black curve), 3D-TPP-COF (red curve) and TPP (blue curve).

Reviewer #3:

***Comment 1:** This manuscript reported a study of a 3D COF and its potential separation for ethylene, and this beautiful work was done through efficient cooperation by three experts in the field of MOF/COF and crystal engineering. The crystalline structure for 3D TPP COF was clearly resolved by cRED technique, remarkably, 3D-TPP-COF exhibited high C₂H₄/C₂H₆ capacity and C₂H₆/C₂H₄ selectivity. I would support the publication of this manuscript in Nature Communications.*

Response: We thank the reviewer for these very positive comments and supporting on the publication.

***Comment 2:** There are a few minor revisions and comments that I would encourage the authors to address before publication.*

1: As far as I am aware of, most COFs show yellow or other obvious color, it is interesting that TPP COF is white in color, authors are suggested to discuss why it is white, and better provide the picture of TPP COF, and relevant spectral characterization like UV or PL can be also included in revision.

Response: We thank the reviewer for bringing this point to our attention. We checked the color of 3D-TPP-COF again. We found that it is more accurate to describe as pale yellow from the powder picture and solid-state UV-vis spectrum of 3D-TPP-COF. Accordingly, we have revised the description in the manuscript and added the picture and solid-state UV-vis spectrum of 3D-TPP-COF into the revised supplementary information as Supplementary Figure 4.

Supplementary Figure 4 Solid-state UV-vis spectrum of 3D-TPP-COF (inset: picture of 3D-TPP-COF)

Comment 3: 2: CIF files are suggested to be deposited into CCDC, which will be useful to the community, especially those who do not have the Materials Studio package.

Response: We thank the reviewer for bringing this point to our attention. We have uploaded the CIF of 3D-TPP-COF to the CCDC database with deposition number 2300940.

Comment 4: 3: It would be helpful to make comparisons of Q_{st} and adsorption selectivity for C_2H_6 and C_2H_4 toward other COFs/MOFs, a table of the comparison results can be listed in ESI to present an intuitive data visualization.

Response: We thank the reviewer for bringing this point to our attention. We have added the following table (Supplementary Table 2) for comparison in the revised supplementary information.

Supplementary Table 2 Summary of reported materials for C_2H_6/C_2H_4 separation.

Materials	T (K)	C_2H_4		C_2H_6		IAST Selectivity (C_2H_6/C_2H_4 , 1/1)	Ref.
		Uptake (mmol/g)	Q_{st} (kJ/mol)	Uptake (mmol/g)	Q_{st} (kJ/mol)		
Zn-FBA	298	1.14	39.8	1.25	42.8	2.9	11
UiO-67-NH ₂	298	4.32	24.5	5.32	26.5	1.7	12
Cu(Qc) ₂	298	0.78	25.4	1.85	29.0	3.4	13
Fe ₂ (O ₂)(dobdc)	298	2.60	36.5	3.32	66.8	4.4	14
PCN-250	298	4.22	21.1	5.21	23.2	1.9	15

MAF-49	298	1.70	48.0	2.70	60.0	1.7	16
MUF-15	293	4.15	28.2	1.96	29.2	4.7	17
COF-1	298	1.92	21.5	2.46	22.5	1.9	18
NKCOF-21	298	3.32	23.6	4.37	26.2	1.6	19
NKCOF-22	298	1.82	24.1	2.94	25.9	1.5	
NKCOF-23	298	2.28	23.0	2.70	24.3	1.3	
3D-TPB-COF-H	293	3.23	25.3	3.25	26.8	1.4	This work
3D-TPP-COF	293	4.70	28.1	4.93	29.7	1.8	This work

Comment 5: 4: BET linear plot should be provided for TPP COF.

Response: We thank the reviewer for bringing this point to our attention. We have added the BET linear plots for both 3D COFs in Supplementary Figure 7.

Supplementary Figure 7 The corresponding nitrogen sorption isotherms (inset: selected linear plots) and pore size distributions of 3D-TPP-COF (a, b) and 3D-TPB-COF-H (c, d).

Comment 6: 5: Typo and format. e.g. a needless letter d in the first page (behind the Email address) should be removed.

Response: We thank the reviewer for pointing out our mistake. We have removed the needless letter in the revised manuscript.

REVIEWERS' COMMENTS

Reviewer #1 (Remarks to the Author):

The authors have satisfactorily addressed all the comments from the reviewers and in particular the results from the additional experiments during the revision stage have made the work further improved. This reviewer recommends its acceptance in the current form.

Reviewer #3 (Remarks to the Author):

[Note from the Editor: Reviewer #3 was asked to look also over the response given to Reviewer #2]

The authors have adressed all my concerns,and from my point of view, the technical points raised from the previous review have also been substantially addressed,I have no further comments at this stage,the revised manuscript can be accepted for publication in its current form.

Point-By-Point Response to the Reviewers' Comments

Reviewer #1:

Comment 1: The authors have satisfactorily addressed all the comments from the reviewers and in particular the results from the additional experiments during the revision stage have made the work further improved. This reviewer recommends its acceptance in the current form.

Response: We thank the reviewer for these very positive comments and supporting on the publication.

Reviewer #3:

Comment 1: The authors have addressed all my concerns, and from my point of view, the technical points raised from the previous review have also been substantially addressed. I have no further comments at this stage, the revised manuscript can be accepted for publication in its current form.

Response: We thank the reviewer for these very positive comments and supporting on the publication.